# The Influence of State Authenticity on the Sense of Meaning in Life of Middle School Students: Evidence from a Daily Diary Investigation and an Authenticity Enhancement Experiment

**DOI:** 10.3390/bs14070550

**Published:** 2024-06-28

**Authors:** Shan Yan, Weihai Tang

**Affiliations:** Faculty of Psychology, Tianjin Normal University, Tianjin 300387, China; 1901030033@stu.tjnu.edu.cn

**Keywords:** SA, MIL, daily diary method, enhance experiment

## Abstract

Authenticity is a positive force for adolescent development. Taking middle school students as the main research objects, this paper examines the causal mechanism between state authenticity (SA) and sense of meaning in life through two studies: the diary method and authenticity level increase experiment. In study 1, through the daily diary data tracking investigation of 130 middle school students in daily life situations, the basic psychological needs (BPNs) and satisfaction with life (SWL) play a chain mediating role between state authenticity and sense of daily meaning in life. State authenticity has a one-way prediction effect on sense of meaning in life. In study 2, 140 participants were randomly divided into two groups (experimental group and control group). Middle school students in the experimental group were activated to recall the authenticity experience events to enhance the level of personal authenticity, and it was found that the subjects reported a higher sense of meaning in life. This study reveals the causal mechanism between authenticity and sense of meaning of life, which has positive practical significance for exploring ways to enhance the sense of meaning of life of middle school students.

## 1. Introduction

Who is the real me? Am I true to myself? As a foundation for human growth and healthy psychological development [1], authenticity and its involvement in the positive development of the individual have been largely ignored by researchers. Adolescence is a time of searching for authenticity, and the sense of being one’s true self is the most important concern. Middle school is a crucial stage of life development, and middle school students are more sensitive. They feel more determined and self-aware. Thinking about the meaning of life has entered a new stage. In this period, middle school students may constantly clarify the goal of life, pursue the meaning of life, or feel empty and meaningless, and the goal is more unclear. The authentic self may provide them with a unique philosophy of life to look at relationships as well as important events, life goals, etc. [2]. People who feel authentic experience a fuller and richer life [3].

Previous studies have focused on the function and significance of the sense of meaning of life, but the formation mechanism of its antecedent variables needs to be further explored. Authenticity as a positive quality in participation in youth development has been largely ignored by research on adolescent psychology. With the rise of positive psychology, the theory and practice of life education with the theme of “calling for life and caring for life” is developing rapidly around the world. It advocates to treat everyone’s potential from a perspective of appreciation, adhere to a positive evaluation orientation of human nature, study how people can better develop and live, and stimulate their inherent constructive strength and positive qualities. Let people learn to share happiness, create happiness, make life more meaningful. Therefore, from the perspective of cultivating positive personal qualities, there are insufficient studies on middle school students’ sense of meaning of life, which need to be further explored. In addition, studies on the sense of meaning in life need to be further explored from the perspective of daily life, and the situation of daily fluctuations and stability have not received sufficient attention. The advantages of daily investigation methods lie in the following: They allow for mental activity to be analyzed within a specific time frame and in the natural context in which it occurs [4]. Therefore, in order to understand the inner and psychological state of middle school students and promote their positive development, we can explore the relationship between the state authenticity of middle school students’ daily life and their sense of meaning in life.

Authenticity refers to people’s subjective feelings of knowing and expressing their true selves, i.e., who they think they are reflects the judgment that a person is acting according to his true self-concept [5,6]. Wood proposed three dimensions of authenticity: self-alienation, authentic living, and accepting external influence [7]. “Self-alienation” refers to the subjective experience of not knowing who you are, not being able to truly understand your own nature, and being detached from your true self. “Real life” refers to the consistency of an individual’s consciousness, belief, and behavior expression, representing that an individual believes that he or she can express and act in accordance with his or her own beliefs and values in various situations. “Acceptance of external influences” refers to the degree to which an individual accepts the ideas of others and feels obligated to conform to their expectations. Generally speaking, individuals with high authenticity have a low sense of self-alienation, are able to practice real life, are less affected by external influences, and will not distort their self-cognition and hinder their true expression and behavior due to external pressure [8]. Authenticity can be further divided into trait authenticity and state authenticity. Unlike the stability and continuity of trait authenticity, state authenticity is a transient experience that can occur anytime, anywhere. It will change due to changes in the external environment, emphasizing a sense of the present [9] and the immediate [10]. In other words, one’s behavior is consistent with one’s core values and abilities at a certain moment, resulting in an emotional experience of becoming one’s true self [11].

A growing number of studies have proved that authenticity is positively correlated with many positive indicators of mental health, such as life satisfaction, subjective well-being, positive emotions, self-esteem, etc. [12,13,14]. People like and value authentic self-concepts more than other types of self-concepts. Studies have shown that the real self has a significant predictive effect on mental health: individuals with a high level of authenticity have higher positive mental health indicators, while they have lower negative mental health indicators, such as less anxiety and depression [15]. Some studies have confirmed that the real self can meet the basic psychological needs of individuals [16], especially the needs of individuals for autonomy and promote people’s sense of well-being based on self-realization and meaning [17]. The higher the level of satisfaction of basic psychological needs, the more conducive to the improvement in happiness and life satisfaction [18]. Individuals whose basic psychological needs are fully satisfied will have more happiness and sense of value, as well as a better psychological state [19]; make independent behavioral decisions according to their own wishes, emotions, values, etc.; maintain a healthy relationship with the world; have competent ability; and realize self-worth. Existing studies have confirmed that authenticity is positively correlated with life satisfaction [5,20,21], and the presentation of an individual’s real self has a positive impact on life satisfaction [22]. The higher the level of authenticity of an individual, the higher the level of life satisfaction and mental health [23]. Life satisfaction is not only a reflection of an individual’s acquired mental state but also has an impact on individual behavior and mental state. Compared with the positive self, the presentation of the true self has a better effect on improving life satisfaction [24].

One of the functions of authenticity is to create meaning for people’s lives. People use their true self-concept as a guide for various important decisions, thus enhancing the meaning of life direction. Choices that are consistent with their true self-concept may be evaluated as more meaningful life pursuits. Studies have shown that authenticity is positively correlated with a sense of meaning in life (MIL), but there is a lack of causal evidence between the two [9]. Research has shown that the impact on the sense of meaning in life depends on people’s attitudes towards their true selves [2]. People who feel authentic experience a fuller and richer life [3]. Authenticity predicts the existence of greater meaning between people in life [25]. Li believes that the real self can play a positive role in promoting middle school students’ sense of meaning in life by meeting their basic psychological needs and improving their self-esteem [26]. However, high life satisfaction will bring individuals a higher sense of meaning in life, and studies have found that college students’ life satisfaction can positively predict their sense of meaning in life [27].

To sum up, authenticity can meet the basic psychological needs of individuals, achieve a satisfactory state of life, realize the sense of value, feel the value of life existence, and stimulate the motivation to pursue life goals. Therefore, it can be inferred that authenticity affects the sense of meaning of life through basic psychological demands and life satisfaction. Therefore, we propose the following hypothesis, and the hypothesis model is shown in Figure 1:

**Hypothesis** **1.**
*Basic psychological needs and life satisfaction play a chain mediating role between state authenticity and sense of meaning in life.*


Compared with trait reality, state reality reflects the real self-differences of individuals in different environments and roles. State authenticity varies from person to person. State authenticity reflects the real feelings of an individual in daily life, which can further and comprehensively reflect the relationship between authenticity and sense of meaning in life. Some longitudinal studies have further explored the relationship between state authenticity and sense of meaning in life. When people feel real, they will improve themselves [28]. Self-improvement can bring the satisfaction of basic psychological needs and happiness and improve mental health. Activating the true self can enhance the meaning of life [29]. When specifically exploring the relationship between the internal three dimensions of authenticity and sense of meaning in life, the study found that authentic life predicted a higher level of sense of meaning in life, while self-alienation and acceptance of external influences predicted a lower level of sense of meaning in life [30]. In an experimental study on authenticity, Guenther found that the authenticity-enhanced group reported a higher sense of meaning in life through enhanced manipulation of authenticity levels [31]. In summary, Hypothesis 2 and Hypothesis 3 are proposed, and the hypothesis model is shown in Figure 2.

**Hypothesis** **2.**
*The authenticity of the previous day’s state predicts the meaning of life the following day.*


**Hypothesis** **3.**
*Under experimental conditions with enhanced levels of authenticity, middle school students will report higher levels of meaning in life.*


The object of this study is middle school students, because the authentic experience is especially important in the group of teenagers aged 12–17, who attach great importance to the experience of the real self in life. Most of the previous studies on the relationship between the true self and the sense of meaning in life were cross-sectional studies, so the causal prediction relationship between the two could not be revealed. This study attempts to solve this problem through two studies: the daily diary method and authenticity experiment.

## 2. Study 1: The Influence of State Authenticity on the Sense of Meaning in Life of Middle School Students—Evidence from Daily Diary Method

In this study, the daily diary method was used to further investigate the relationship between state authenticity and sense of meaning in life of middle school students in the environment of daily natural life. Compared with traditional research methods, real-time assessment in real situations of daily life can reduce individual recall bias, increase ecological validity, and make short-term causal prediction. This study focuses on the evaluation of internal mental state changes.

### 2.1. Methods and Measures

#### 2.1.1. Participants

Previous studies suggested that for inter-individual predictors, the sample size should be at least 50 people, and for intra-individual predictors, the measurement time point should be at least 5 days [32]. Calculations using G*Power show that a statistical test power of 0.80 (α = 0.05) would be required for at least 120 subjects at 7 days of follow-up with an in-group effect (γ10.std = 0.10, ICC = 0.50). In a middle school in Tianjin, 140 high school students were recruited to conduct a diary survey for 7 consecutive days. Among them, 10 subjects did not complete the survey tasks as required, and finally 130 subjects (58 males, 72 females) were valid, with an average age of 15.72 ± 1.68 years old. The sample size used in this study meets the requirements.

#### 2.1.2. Procedure

This study was approved by the research ethics committee of the university of the authors. First, participants were given brief instructions and requirements for filling out questionnaires. Then, the participants were asked to complete a short (about 8 min) online questionnaire, including a series of questions at the end of the day (before 11 pm every day). At the same time, this study obtained the informed consent of the school and parents and also obtained the informed consent of the participants themselves. The students were paid CNY 20 for completing the full 7-day survey.

#### 2.1.3. Measures

##### Daily State Authenticity Questionnaire (DSAQ)

Participants’ daily state authenticity was measured. The items of the authenticity questionnaire were adapted to measure the participants’ daily authenticity level. The authenticity dimension of life items used the phrases “Today, I stand by what I believe” and “Today, I am true in most situations”. The self-distancing dimension project used “Today, I feel as if I don’t know myself very well” and “Today, I feel disconnected from the ‘real me’”. The accepting of outside influences dimension items used “Today I did what I was told to do” and “Today other people influenced me a lot”. Each item is rated on a 7-point scale, (1 = strongly disagree, 7 = strongly agree). In this study, the α coefficients of this questionnaire were 0.774 and 0.871 within and between subjects, respectively.

##### Daily Meaning in Life Questionnaire (DMILQ)

The meaning in daily life was measured, and the items in the meaning of life questionnaire were adapted to measure the level of participants’ daily sense of meaning in life. Among them, two questions were used in the existential dimension of sense of meaning of life, “There is no clear purpose in my life today” and “I found a satisfying purpose in my life today”. The dimension of the sense of meaning of life uses two questions: “Today I am looking for a purpose or mission in my life” and “today I am looking for something that makes me feel that my life is meaningful”. Each item was rated on a 7-point scale from 1 (strongly disagree) to 7 (strongly agree). In the present study, the α coefficient of this questionnaire was 0.780 and 0.673 within and between subjects, respectively.

##### Daily Basic Psychological Needs Questionnaire (DBPNQ)

For the measurement of daily basic psychological needs, the modified questionnaire is used to measure the daily basic psychological needs of the subjects. Items with three dimensions were included. The autonomy needs included two questions: “I am happy to express my own ideas and opinions today” and “I can decide my own affairs today”. The ability need dimension included two questions, “I have done a good job today” and “I can do interesting things or learn new skills today”. The relationship needs dimension included “I got along well with everyone I came into contact with today” and “People were friendly to me today”. Each item is rated on a 7-point scale from 1 (strongly disagree) to 7 (strongly agree). In the present study, the α coefficient of this questionnaire was 0.818 and 0.935 within and between subjects, respectively.

##### Daily Life Satisfaction Questionnaire (DLSQ)

The currently frequently used two items [33] are used in studies to assess daily life satisfaction. This requires subjects to rate the first item, “How was your day?” on a 7-point scale (1 = bad, 7 = excellent), and then they rate the second item, “Are you satisfied with your life today?” (1 = very dissatisfied, 7 = very satisfied). These items have shown good reliability and validity in previous daily diary studies [34]. In the present study, the α coefficients of the questionnaire were 0.899 and 0.956 within and between subjects, respectively.

#### 2.1.4. Data Analysis

##### Common Method Variance

In order to avoid common method bias effect, the Harman single factor test was conducted to collect data mainly in the form of self-report of research subjects. The results showed that there were four eigenvalues greater than one factor number, and the variance explained by the first factor was 37.1%, lower than the critical value of 40%, indicating that there was no significant common method bias in this study.

##### Statistical Analysis

The data obtained through the diary method of investigation constitute a multi-level structure with occasions (days) nested within the individual. To evaluate the within-subject mediation model, we return everyday authenticity to a sense of meaning in everyday life.

To test the hypotheses presented in this study, we used robust maximum likelihood estimates in Mplus8.3 [35], analyzed using the multistage structural equation modeling method MSEM [36,37]. We used a maximum likelihood estimate with robust standard error (MLR) to evaluate the model parameters. For model fit evaluation, we used the comparative fit index (CFI), Tucker–Lewis index (TLI), approximate root-mean-square error (RMSEA), and standardized root-mean-square residual (SRMR). CFI and TLI values >0.95 are considered to indicate an excellent fit [38]), while RMSEA values <0.06 and SRMR values <0.08 are considered to indicate a good fit [39].

In addition, because Bayesian estimation has many advantages in multistage analysis (e.g., suitable for testing complex models with small cluster sizes, negative estimation of variance terms that occur relatively frequently in multistage analysis based on ML estimators is avoided [40], we use Bayesian estimators to test the final MSEM model as well as the hypothesized indirect effects. In order to prove the mediating effect, we calculate the indirect effect in the proposed mediation chain and test its statistical significance. MSEM methods are particularly useful for data analysis based on dense longitudinal methodology [41,42]. MSEM successfully decomposes the within-subject and between-subject variances in potential variables, enabling various hypothetical relationships to be specified and tested at different levels of analysis.

In this study, the mediation chain of the hypothesis can be annotated as 1-1-1-1. In order to determine the statistical significance of the hypothesized indirect effect, given the asymmetric distribution of the product coefficients, we used a Bayesian estimator and the MODEL CONSTRAINT option in M plus to obtain a 95% confidence interval (CI) for the indirect effect [36].

### 2.2. Results

#### 2.2.1. Descriptive Statistical Analysis

This study used the daily diary method to investigate the daily changes in individual psychological status. The results in Table 1 show the average values of individuals’ daily state authenticity, sense of meaning in life, basic psychological needs, life satisfaction, intra-subject and inter-subject variances, intra-group correlation coefficient (ICC), and intra-subject and inter-subject correlation coefficient between variables.

#### 2.2.2. Intermediary Effect Analysis

To further examine the relationship between authenticity and sense of meaning of life, we construct a multilevel structural equation model in which authenticity predicts sense of meaning in life. According to the results of correlation analysis, multi-level analysis is suitable. The results show that the model fits well: χ^2^_(1)_ = 0.789, CFI = 1.000, TLI = 1.004, SRMR_WITHIN_ = 0.000, SRMR_BETWEEN_ = 0.024, RMSEA = 0.000.

As in Table 2, the results within individuals show that state authenticity can significantly positively predict basic psychological needs, life satisfaction, and meaning in life, while basic psychological needs can significantly positively predict life satisfaction and meaning in life, and life satisfaction can significantly positively predict meaning in life. The results between individuals show that after controlling for gender, state authenticity can significantly positively predict basic psychological needs and life satisfaction, basic psychological needs can significantly positively predict life satisfaction, and basic psychological needs can significantly positively predict meaning in life.

Bayesian estimates were used to calculate the chain mediating effect of basic psychological needs and life satisfaction between authenticity and sense of meaning in life within individuals. As shown in Table 3, the reality of state within individuals can influence the sense of meaning of life through the mediating effect of basic psychological needs and the chain mediating effect of basic psychological needs and satisfaction. The state authenticity between individuals affects the sense of meaning of life through the mediating effect of basic psychological needs, and there is no chain mediating effect.

Therefore, there exists a chain mediating effect between the basic psychological needs and life satisfaction in individuals and the sense of state authenticity and meaning of daily life (Figure 3). The basic psychological needs of individuals play a mediating role between state authenticity and sense of daily meaning in life (Figure 4).

#### 2.2.3. Multi-level Hysteresis Effect Analysis

As shown in Table 4, according to the multi-level cross-lag effect path analysis results, the level of state authenticity one day before can predict the level of sense of meaning in life one day after, while the level of sense of meaning in life one day before cannot significantly predict the level of state authenticity one day after. It can be seen that during the week of the diary survey, authenticity and sense of meaning of life showed high intra-individual stability within a short period of 7 days. This indicates that if an individual presents a high level of state authenticity and sense of meaning in life, it will continue to a certain extent. Therefore, state authenticity has a one-way predictive effect on sense of meaning in life, as in Figure 5.

## 3. Study 2: The Impact of State Authenticity on Middle School Students’ Sense of Meaning in Life—Evidence from Experiments on Enhancement of Authenticity Level

To conduct a two-level inter-subject experiment, we assigned subjects to an experimental group and a control group under an authenticity or control condition. We manipulated authenticity under a recall situation and tested its effect on the sense of meaning in life.

### 3.1. Methods and Measures

#### 3.1.1. Participants

We investigated the influence of enhanced authenticity level on the sense of meaning in life under experimental conditions. G*power 3.1 analysis showed that the main effect of enhanced authenticity level was 0.5 when the significance level (α) was less than or equal to 0.05 and the statistical testing power was (1 − β) 0.8 with 64 observation samples in each group. Therefore, 140 high school students were investigated in this study. The participants obtained informed consent from the school and their parents and provided their own consent (the consent rate of participants = 100%). The participants were randomly divided into an experimental (real) group of 70 and a control group of 70, and they completed the experiment in their regular classrooms. They completed the work independently and returned it in a file bag when they were finished. They guided the experiment procedure by interpreting instructions from the subject and by written instructions.

#### 3.1.2. Procedure

This study was approved by the research ethics committee of the university of the authors. The procedure was adapted from the standard procedure used by Kifer [43], in which both the real group and the control group were asked to first complete measurements of the authenticity level and the sense of meaning in life, as in Figure 6. Next, the real group members were asked to recall “a specific event in which you experienced your own behavior being consistent with your inner authenticity or experiencing a sense of authenticity”. Participants were then asked to relive the event in their imaginary recollection and write a short essay of no less than 100 words describing exactly what happened during the event and how it made them feel. Participants also reported when the event occurred (pre-experiment M_day_ = 247, SD = 280). The process took 10–20 min to complete. Next, the participants’ level of sense of meaning in life was measured using the State Authenticity Questionnaire and the Sense of Meaning in Life questionnaire.

The control group was required to complete a record and recall of the previous night’s events, and then fill in the state authenticity questionnaire and the sense of meaning in life questionnaire again.

#### 3.1.3. Measures

##### Authenticity Questionnaire (AQ)

The state authenticity measurement that the subjects need to complete is adapted from the authenticity scale. Before each item of the original measurement questionnaire, we add the “Thinking about this situation makes me feel…”, for example, “Thinking about this situation makes me feel better about being myself than being popular”, or “Thinking about this situation makes me feel like I am who I really am”. The scale ranges from 1 = strongly disagree to 7 = strongly agree [43]. We calculated the average score of the responses for each item.

##### Meaning in Life Questionnaire (MIL)

The participants completed the Meaning of Life Questionnaire [44], which was also adapted. Each item on the original measurement questionnaire was preceded by the phrase “Thinking about this situation makes me feel…”, for example, “Thinking about this situation makes me feel that I now understand the meaning of my life” (1 = strongly disagree, 7 = strongly agree). We calculate the average score for each item’s rating.

#### 3.1.4. Manipulation Check

As an operational check, four graduate students read the participants’ writing and independently coded their level of compliance with the instructions on a scale of 0 = poor to 2 = good. Fidelity was considered adequate when at least three of the four encoders judged instruction compliance as “good” and none judged instruction compliance as “bad”. Six participants (8.6%) had works that did not meet the fidelity criteria. We excluded these students from the analysis, and the final experimental group underwent data analysis with a sample N = 64. The excluded participants did not differ from the included participants in any of the study variables.

### 3.2. Results

#### 3.2.1. Validity Check of Enhancement Experiment

The independent sample *t* test was adopted, as shown in Table 5. There was no significant difference in the authenticity level of the pre-test between the experimental group and the control group, which also indicated that the pre-test of the two groups was homogeneous. Further experimental studies can be conducted. There was no significant difference between the authenticity level measured by the control group after recalling the events of the previous night and the authenticity level measured before.

There was no significant difference between the control group’s levels of authenticity measured after recalling the previous night’s events and those measured before, as shown in Table 6. In the post-test of the authenticity enhancement experiment, there were significant differences between the experimental group and the control group in authenticity enhancement (t = 2.15, *p* < 0.05, d = 0.38), indicating that the manipulation of authenticity enhancement in this experiment was successful, and this experiment was effective in enhancing the authenticity of the experimental group, as shown in Table 7.

#### 3.2.2. Analysis of the Enhancement of the Authenticity

In the experimental group, there were significant differences between the authenticity level measured after recalling the events that reported the authenticity experience or feeling and the authenticity level measured before the enhancement experiment (t = −2.452, *p* < 0.05, d = 0.43), as shown in Table 8. It can be seen from Figure 7 and Figure 8 that the authenticity level of participants in the two groups changed significantly before and after the experiment.

Further, we want to know how the internal three-dimension data of the experimental group changed before and after the enhancement of the authenticity level. The results are shown in Table 8, where there is no significant difference in the self-alienation dimension before and after the enhancement, there is a significant difference in the acceptance of external influence dimension, and there is also a significant difference in the authenticity dimension. Therefore, this authenticity enhancement experiment can improve the level of individual authenticity, especially the dimension of authentic life, followed by the acceptance of external influences.

#### 3.2.3. Comparison of Sense of Meaning in Life

In Table 9, the analysis of the pre-test data with enhanced authenticity level shows that there is no significant difference between the experimental group and the control group in the level of sense of meaning of life and the internal dimensions of sense of meaning of life, indicating that the two groups of subjects in the pre-test are homogenous.

Next, we can further investigate whether there are significant differences in the sense of meaning of life before and after the experiment, and whether the sense of meaning of life is improved under the condition of increased authenticity level.

As can be seen from Table 10, there were significant differences between the experimental groups in sense of existential meaning (*p* < 0.001) and sense of meaning seeking (*p* < 0.01) before and after the authenticity enhancement experiment. With the enhancement of authenticity, the sense of meaning in life will be improved significantly. Under the condition of enhancing authenticity, the improvement level of meaning existence is more significant. The changes in the sense of meaning in life of the two groups before and after the experiment are shown in Figure 9.

## 4. Discussion

The contribution of this study is the investigation of the causal mechanism between authenticity and sense of meaning in life in middle school students. Middle school students are in the youth development period, with the need for self. They begin to pay attention to self-evaluation and improvement. During this period, students have great changes in physical and psychological aspects, and their self-awareness is enhanced. Many teens believe that they will perform best if they can successfully “take off the mask” and “be themselves” [1]. In study 1, through the daily diary data tracking survey of middle school students in daily life situations, the results show that authenticity has a positive predictive effect on sense of meaning in life, and the basic psychological needs and life satisfaction within individuals have a chain intermediary effect between state authenticity and sense of daily meaning in life, indicating the impact of changes in authenticity within individuals. There is a mediating effect between the state authenticity and the sense of daily meaning in life, which indicates that there are differences between individuals. It was found that the level of state authenticity the day before predicted the level of sense of meaning the next day. The state of real life is usually a positive experience. Consistent with the results of previous studies, the authenticity helps to improve the basic psychological needs of individuals [45] and the sense of meaning in life [11], and the state authenticity is more likely to meet the psychological needs of individuals than the state inauthenticity [46]. This conclusion also exists in the group of middle school students. Deci proposed that when individuals act according to their own will, they show autonomy and experience satisfaction, success, and meaning through independent choice [47]. Authentic experience helps to improve the perception of the meaning in life, and when individuals make behavioral choices relying on their real self, they can better reflect the pursuit of life goals and values [48]. Basic psychological needs and life satisfaction have a chain mediating effect between authenticity and sense of life meaning. Thomaes argues that real people have a sense of completeness and self-coherence [49]. This sense of completeness and self-cohesion promotes satisfaction. Middle school students are in adolescence, they have a strong need for their real selves, and they hope to be able to do what they like without being controlled. Authentic experience is close to self-feeling in daily life, and the pursuit of authentic behavior can help them fully stimulate their internal motivation, so as to obtain the value of their existence and continuously improve, so as to meet their basic psychological needs, bring a sense of accomplishment and satisfaction, and enhance the perception of the meaning of daily life. The results of this study reveal an internal path, that is, state authenticity can make individuals feel satisfied and valuable through the bridge of basic psychological needs, so that individuals can obtain a stronger life existence that is more valuable and meaningful. The results of study 1 also show that in the daily life environment, students’ sense of authenticity experience changes in different situations after learning life events, and this change will bring changes in the sense of meaning in life. At the same time, we found that in a short period of time, individuals’ state authenticity and sense of life meaning are relatively stable, and individuals with a high level of authenticity also have a high level of sense of meaning in life, and vice versa.

In addition, when we further investigate the change in the internal dimension of authenticity, we find that the level of authentic living has been significantly improved, followed by the acceptance of external influences, while the level of self-alienation has no significant change. This finding is consistent with conclusion that fulfillment of meaning in life is associated with a more authentic life [45]. The reason may be that when individuals recall the relevant events of authentic experience, they mostly record the relevant events from the perspective of action. The authentic experience that students can understand is mostly related to their own behavior, and authentic behavior may be an important way for middle school students to experience the sense of meaning. Authentic living is a feeling of self-relaxation and positive emotional experience, which contributes to better self-cognition and ideal self-experience [45], and thus has a stronger impact on the sense of meaning in life. The dimension of self-alienation is embodied in an individual’s self-awareness, whether he knows who he is or not, which is formed through long-term self-cognition. As middle school students, the ability to view their own thinking in this way is probably rare, and this experience cannot be improved in the short term by experiments with immediate recall of events.

Furthermore, the results reveal the causal relationship between state authenticity and sense of meaning in life. Study 2 examines the effect of state authenticity on the sense of meaning in life through a research paradigm in laboratory conditions. The results show that the manipulation of the experiment is effective. This method can improve the individual’s immediate sense of real experience and thus enhance the immediate sense of meaning in life. Consistent with previous findings, authentic living predicted higher levels of sense of meaning in life [30]. This increase in authenticity can be seen as a kind of self-improvement, where the recall of past events of authenticity increases the individual’s reflection and thus enhances the sense of meaning in life [31].

The experiment of enhancing the authenticity level has a significant impact on the improvement of middle school students’ sense of existence of life meaning, indicating that this experience can make them realize the value of the existence of life through memory and reflection, and has a positive impact on the current cognitive experience of life. However, this kind of experience and feeling is not closely related to the pursuit and transformation of life goals.

In summary, this study aims to investigate the mechanism of authenticity’s influence on the sense of meaning in life, expand the research perspective on the sense of meaning in life, and provide a new entry point for the intervention to enhance the sense of meaning in life. Measures should be taken to help middle school students find their true self, understand themselves, and gain a more meaningful, positive, and healthy life.

There are some limitations to this study that could be improved in the future. All the instrument scales used in this study have good reliability and validity, but there may be some bias in the form of self-report. In this paper, daily survey tracking using the diary method is used for short-term prediction, and the effect of long-term longitudinal tracking can be further investigated. In the future, authenticity can be taken as the entry point to further intervention programs to enhance the sense of meaning in life of middle school students, effectively improve their self-value recognition and pursuit of meaning in life, and maintain a positive mental health state.

## 5. Conclusions

The results show that there is a correlation between state authenticity and sense of meaning in life and between basic psychological needs and the life satisfaction of middle school students in daily life. Basic psychological needs and life satisfaction play a chain intermediary role between state authenticity and sense of meaning in life. The authenticity level of the previous day can predict the level of the sense of meaning in life of the next day, and the state authenticity has a one-way prediction effect on the sense of meaning in life. As middle school students’ levels of authenticity rise, so does their level of meaning in life. Among them, the sense of existence meaning increased significantly.

## Figures and Tables

**Figure 1 behavsci-14-00550-f001:**
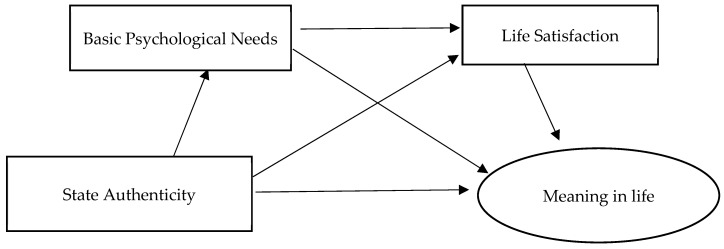
Hypothesis model of mediation effect.

**Figure 2 behavsci-14-00550-f002:**
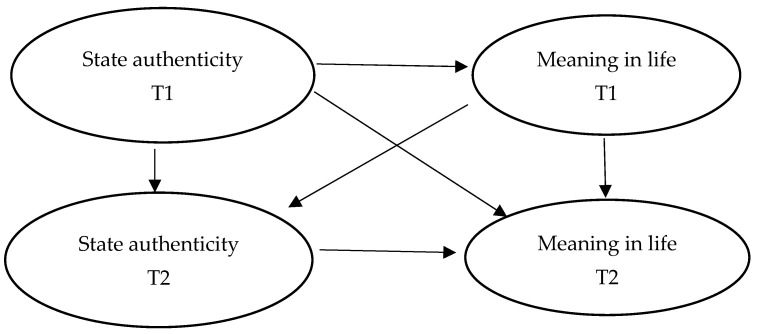
Cross hysteresis effect hypothesis model.

**Figure 3 behavsci-14-00550-f003:**
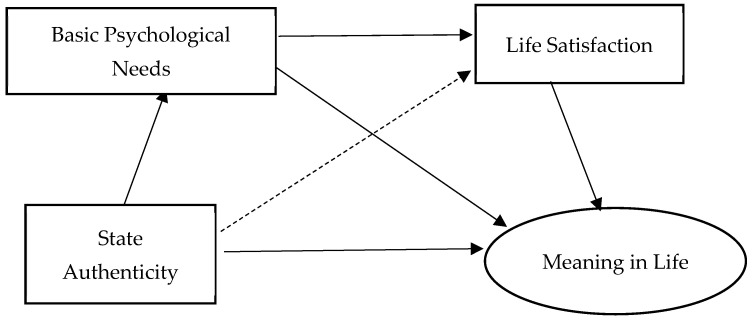
Chain mediation effect model (within).

**Figure 4 behavsci-14-00550-f004:**
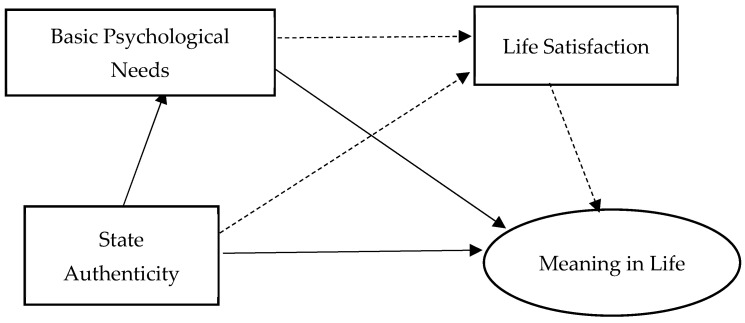
Chain mediation effect model (between).

**Figure 5 behavsci-14-00550-f005:**
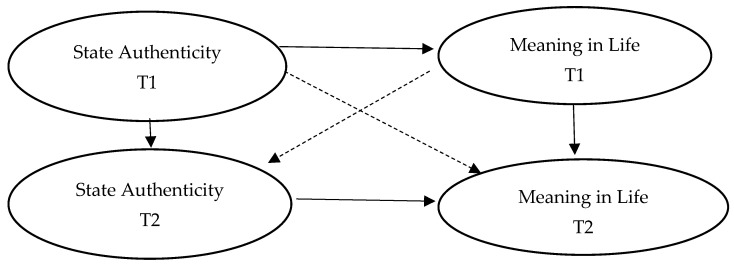
The predictive effect (within).

**Figure 6 behavsci-14-00550-f006:**
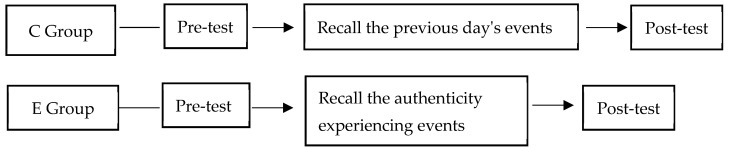
Authenticity enhancement procedure.

**Figure 7 behavsci-14-00550-f007:**
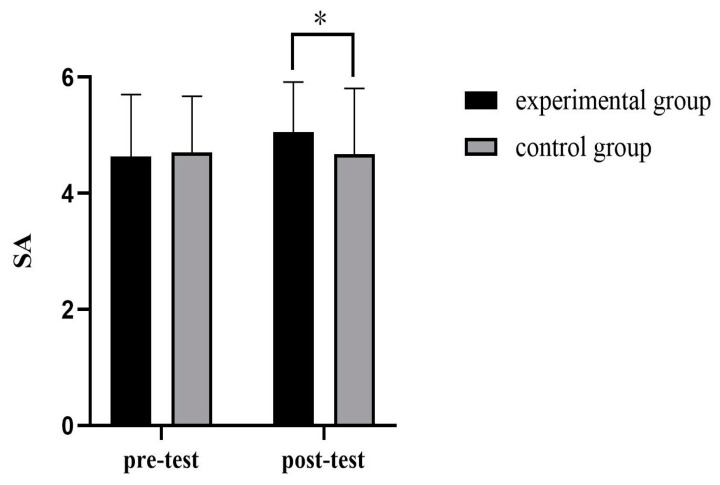
Comparison of state authenticity between two groups. * *p* < 0.05.

**Figure 8 behavsci-14-00550-f008:**
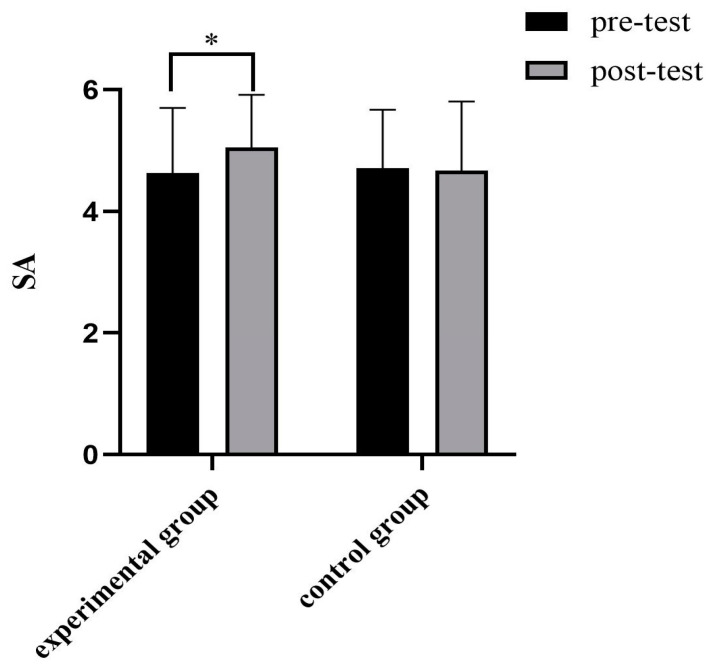
Comparison of state authenticity in experiment group. * *p* < 0.05.

**Figure 9 behavsci-14-00550-f009:**
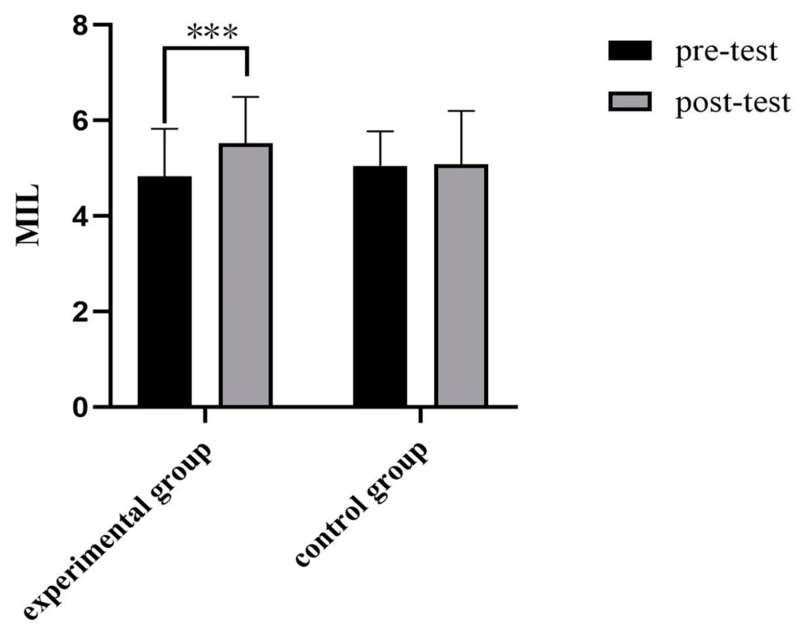
Comparison of meaning in life in experiment group. *** *p* < 0.001.

**Table 1 behavsci-14-00550-t001:** Descriptive statistics and intra-group correlation coefficient (ICC).

	*M*	*SD_WITHIN_*	*SD_BETWEEN_*	*ICC*	Authenticity	Sense of Meaning in Life	Basic Psychological Needs	Life Satisfaction
Authenticity	29.76	4.703	5.761	0.611		0.421 **	0.342 **	0.361 ***
Sense of meaning in life	16.75	3.079	3.950	0.543	0.521 ***		0.739 ***	0.600 ***
Basic psychological needs	33.99	5.103	6.132	0.641	0.388 ***	0.674 ***		0.561 ***
Life satisfaction	11.01	1.524	2.189	0.399	0.415 ***	0.579 ***	0.572 ***	

** *p* < 0.01 *** *p* < 0.001.

**Table 2 behavsci-14-00550-t002:** Multi-level structural equation model.

Models	Within			Between		
	Estimate	SD	*p*	Estimate	SD	*p*
Control Variable						
Gender → Sense of meaning in life				0.072	0.342	0.834
Gender → Basic psychological needs				0.716	0.841	0.394
Gender → Life satisfaction				0.238	0.220	0.279
Direct Effects						
Authenticity → Sense of meaning in life	0.380	0.026	<0.001	0.076	0.045	0.834
Authenticity → Basic psychological needs	0.498	0.045	<0.001	0.366	0.101	0.002
Authenticity → Life satisfaction	0.126	0.018	<0.001	0.052	0.026	0.043
Basic Psychological Needs → Sense of meaning in life	0.146	0.039	< 0.001	0.344	0.045	<0.001
Life satisfaction → Sense of meaning in life	0.284	0.066	<0.001	0.511	0.206	0.013
Basic Psychological needs → Life satisfaction	0.226	0.021	<0.001	0.143	0.027	<0.001

**Table 3 behavsci-14-00550-t003:** Mediation effect test.

Models	Estimate	Post.sd	*p*	95%CI	
				Lower	Upper
**Within**					
Authenticity → Basic psychological needs → Sense of Meaning in Life	0.073	0.012	<0.001	0.050	0.098
Authenticity → Life Satisfaction → Sense of Meaning in Life	0.035	0.007	0.104	0.022	0.051
Authenticity → Basic Psychological Needs → Life Satisfaction → Sense of Meaning in Life	0.032	0.006	<0.001	0.020	0.045
**Between**					
Authenticity → Basic Psychological Needs → Sense of Meaning in Life	0.125	0.039	<0.001	0.055	0.208
Authenticity → Life Satisfaction → Sense of Meaning in Life	0.025	0.018	0.030	0.001	0.066
Authenticity → Basic Psychological Needs → Life Satisfaction → Sense of Meaning in Life	0.025	0.013	0.002	0.007	0.057

Note: *p* is a single-tail test.

**Table 4 behavsci-14-00550-t004:** Relationship between state authenticity and sense of meaning in life.

Effects	Variable	Beta.	SE	95%CI
Autoregressive Effect	Authenticity_T1_ → Authenticity	0.478 ***	0.069	[0.339, 0.608]
	Sense of meaning in_T1_ life → Sense of meaning in life	0.344 ***	0.071	[0.213, 0.485]
Cross Hysteresis Effect	Authenticity_T1_ → Sense of meaning in life	0.233 ***	0.047	[0.134, 0.321]
	Sense_T1_ of meaning in life → Authenticity	0.065	0.088	[−0.110, 0.228]

Note: T1: the day before, T2: the day after; SE: Standard error; *** *p* < 0.001.

**Table 5 behavsci-14-00550-t005:** Comparison of state authenticity between two groups in pre-test (*M* ± *SD*).

	Experimental Group (n = 64)	Control Group (n = 64)	*t*	*p*	Cohen’s d
State Authenticity	4.61 ± 1.10	4.69 ± 0.95	0.45	0.650	0.07
Self-alienation	4.80 ± 1.21	4.72 ± 1.57	0.33	0.740	0.05
Accepting of external influence	4.10 ± 1.27	4.35 ± 1.22	1.16	0.250	0.20
Authentic life	4.99 ± 1.04	5.05 ± 0.9	0.34	0.736	0.06

**Table 6 behavsci-14-00550-t006:** Comparison of state authenticity in control group (*M* ± *SD*).

	Pre-Test (n = 64)	Post-Test (n = 64)	*t*	*p*	Cohen’s d
State Authenticity	4.69 ± 0.95	4.88 ± 1.00	1.06	0.292	0.19

**Table 7 behavsci-14-00550-t007:** Comparison of state authenticity between two groups in post-test (*M* ± *SD*).

	Pre-Test (n = 64)	Post-Test (n = 64)	*t*	*p*	Cohen’s d
State Authenticity	5.05 ± 0.87	4.67 ± 1.13	2.14	0.034	0.38

**Table 8 behavsci-14-00550-t008:** Comparison of state authenticity in experimental group (*M* ± *SD*).

	Post-Test (n = 64)	Pre-Test (n = 64)	*t*	*p*	Cohen’s d
State Authenticity	5.04 ± 0.87	4.61 ± 1.10	2.45	0.015	0.43
Self-alienation	5.06 ± 1.05	4.80 ± 1.21	−1.30	0.197	0.22
Accepting of External Influences	4.69 ± 0.94	4.10 ± 1.27	−3.02	0.003	0.53
Authentic Life	5.41 ± 0.90	4.99 ± 1.04	−2.41	0.017	0.43

**Table 9 behavsci-14-00550-t009:** Comparison of the sense of meaning in life between two groups in pre-test (*M* ± *SD*).

	Experimental Group (n = 64)	Control Group (n = 64)	*t*	*p*	Cohen’s d
Sense of meaning in Life	4.83 ± 1.00	5.05 ± 0.72	−1.415	0.160	0.25
Sense of meaning existence	4.85 ± 1.08	4.97 ± 1.18	−0.607	0.545	0.11
Sense of meaning seeking	4.81 ± 1.03	5.13 ± 1.09	−1.679	0.096	0.30

**Table 10 behavsci-14-00550-t010:** Comparison of the sense of meaning in life in the experimental group (*M* ± *SD*).

	Post-Test (n = 64)	Pre-Test (n = 64)	*t*	*p*	Cohen’s d
Sense of meaning in life	5.52 ± 0.97	4.83 ± 1.00	3.958	0.000	0.70
Sense of meaning existence	5.77 ± 0.88	4.85 ± 1.08	5.284	0.000	0.93
Sense of meaning seeking	5.28 ± 1.21	4.81 ± 1.03	2.306	0.023	0.41

## Data Availability

The data presented in this study are not available due to privacy.

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
