# Peer review of "The Influence of State Authenticity on the Sense of Meaning in Life of Middle School Students: Evidence from a Daily Diary Investigation and an Authenticity Enhancement Experiment"

_behavsci, 2024, doi:10.3390/bs14070550_

Round 1

Reviewer 1 Report

Comments and Suggestions for Authors

Comments on the Quality of English Language

Minor editing of English language required

Author Response

Dear reviewer:

Thank you very much for your professional commments on this article ! We have made extensive corrections to our previous draft as your suggestions!

Reviewer 2 Report

Comments and Suggestions for Authors

Thank you for the opportunity to review this manuscript. I guess, this is the work of a student, under supervision, of a topic of his/her great interest. 

I find the work problematic due to several issues. I will only outline the major issues, and not go into details. 

1. the concepts are confusing, not well defined and their internal relationships are not spelled out. In the end, I get a feeling that all variables somehow measure the same thing. The theoretical framework is underdeveloped and needs extensive work. References are few, and I miss the "greater story" in the text, reflected in the references.

2.It is nice that the authors try to make causal inference of authenticity predicting meaning of life, in the experimental study. I am not sure that causal inferences can be made within the design of recollecting a memory, that probably felt both authentic and meaningful. 

3. In the introduction, youth is hinted upon as a significant developmental period. This is not further developed. 

4. Again, the discussion lacks a theoretical framework, references and "the big picture". Without them, the study is really only jargon.   

Comments on the Quality of English Language

English per se is ok, however, argumentation is underdeveloped.

Author Response

Thank you very much for your professional commments on this article ! We have made extensive corrections to our previous draft as your suggestions!

Reviewer 3 Report

Comments and Suggestions for Authors

The aim of the paper is to investigate the causal mechanism between state authenticity and sense of meaning of life. In my opinion, the main contributions are related to its 2 studies - 1) diary method and 2) authenticity enhancement experiment. In sum, the authenticity influences the meaning of life through basic needs and life satisfaction. Generally, I think the topic is interesting and also some of the results. However, I do think that your paper needs to be more elaborated to be published. In the abstract you say that you conducted 2 studies, but not the number of participants. I also suggest that you include a conclusion. Results and discussion: Even if there is an interesting discussion, I miss the relation to previous research. In conclusion, I suggest you to relate your findings to previous research. Generally, I would like a better argumentation in relation to previous research in discussion and conclusion. 

Author Response

(The authors gave the same response as above.)
